# Transcranial Magnetic Stimulation Attenuates Dyskinesias and FosB and c-Fos Expression in a Parkinson’s Disease Model

**DOI:** 10.3390/brainsci14121214

**Published:** 2024-11-29

**Authors:** Fernanda Ramírez-López, José Rubén García-Montes, Diana Millán-Aldaco, Marcela Palomero-Rivero, Isaac Túnez-Fiñana, René Drucker-Colín, Gabriel Roldán-Roldán

**Affiliations:** 1Departamento de Neuropatología Molecular, Instituo de Fisiología Celular, Universidad Nacional Autónoma de México, Ciudad de México 04510, Mexico; fer.ramirezm4@gmail.com (F.R.-L.); dmillan@ifc.unam.mx (D.M.-A.); marcelap@ifc.unam.mx (M.P.-R.); drucker@unam.mx (R.D.-C.); 2Instituto Cajal, 28002 Madrid, Spain; joserubengarciamontes@gmail.com; 3Departamento de Bioquímica y Biología Molecular, Facultad de Medicina y Enfermería, Universidad de Córdoba, 14014 Cordoba, Spain; fm2tufii@uco.es; 4Instituto Maimónides de Investigación Biomédica de Córdoba (IMIBIC), 14014 Cordoba, Spain; 5Departamento de Fisiología, Facultad de Medicina, Universidad Nacional Autónoma de México, Ciudad de México 04510, Mexico

**Keywords:** Parkinson’s disease, L-DOPA-induced dyskinesias, dopamine receptors, transcranial magnetic stimulation, motor cortex, striatum

## Abstract

Background/Objectives: Dopamine replacement therapy for Parkinson’s disease (PD) may lead to disabling incontrollable movements known as L-DOPA-induced dyskinesias. Transcranial magnetic stimulation (TMS) has been applied as non-invasive therapy to ameliorate motor symptoms and dyskinesias in PD treatment. Recent studies have shown that TMS-induced motor effects might be related to dopaminergic system modulation. However, the mechanisms underlying these effects of TMS are not fully understood. Objectives: To assess the expression of FosB and c-Fos in dopamine-D1 receptor-containing cells of dyskinetic rats and to analyze the effect of TMS on dyskinetic behavior and its histological marker (FosB). Methods: We investigated the outcome of TMS on cellular activation, using c-Fos immunoreactivity, on D1 receptor-positive (D1R+) cells into the motor cortex and striatum of dyskinetic (*n* = 14) and intact rats (*n* = 14). Additionally, we evaluated the effect of TMS on the dyskinesia global score and its molecular marker, FosB, in the striatum (*n* = 67). Results: TMS reduces c-Fos expression in D1R+cells into the motor cortex and striatum. Moreover, TMS treatment attenuated dyskinesias, along with a low stratal FosB expression. Conclusions: The current study shows that TMS depressed FosB and c-Fos expression in D1R+ cells of the dorsal striatum and motor cortex, in accordance with previous evidence of its capacity to modulate the dopaminergic system, thus suggesting a mechanism by which TMS may mitigate dyskinesias. Additionally, our observations highlight the potential therapeutic effect of TMS on dyskinesias in a PD model.

## 1. Introduction

Dyskinesia is a disabling motor complication that commonly arises from prolonged treatment with the dopamine (DA) precursor, L-DOPA, in Parkinson’s disease (PD). Dyskinesia is characterized by involuntary, often debilitating movements and is a major complication in PD treatment. It typically emerges after extended l-DOPA use and tends to persist even after drug withdrawal, suggesting lasting changes in the brain’s dopamine response. Managing dyskinesia is challenging, with limited effective treatments available. Animal models, including 6-OHDA-lesioned rodents and MPTP-lesioned primates, are essential for investigating its mechanisms and for developing new therapies, as they exhibit dyskinetic features similar to those seen in patients [1,2].

Several studies have suggested a major role for dopamine D1 receptors (D1R) in dyskinesia manifestation [3,4,5]. D1R activation can induce dyskinetic movements similar to those caused by l-DOPA, as shown by D1 agonists like ABT-431 and SKF 82958, which trigger dyskinesias in both humans and animal models [6,7]. Recent studies using optogenetics to activate the striatum in 6-hydroxydopamine rat models have induced dyskinesias [5], while fiber photometry has shown the synchronized overactivity of striatal D1R+ neurons during dyskinetic episodes. Importantly, the optogenetic deactivation of these D1R+ neurons effectively inhibited most dyskinetic behaviors in LID animals [8]. In the dopamine-denervated striatum, the hypersensitivity of these receptors leads to the excessive activation of PKA/DARPP-32, ERK/Elk/MSK1, and mTORC1 signaling pathways [9], followed by an upregulation of immediate early genes inducing FosB [10]. FosB is expressed mainly in D1R-containing neurons [11] and correlates with the severity of the dyskinesia manifestation [10,12]. Moreover, inhibiting FosB expression in specific striatal areas reduces dyskinetic movements [10], justifying its use as a histological marker. Other groups have demonstrated that chronic l-DOPA treatment in 6-OHDA-lesioned rats significantly increases the number of FosB-immunopositive cells, not only in the dorsolateral striatum but also in the cingulate cortex on the lesioned side [13]. In addition to the observed striatal alterations, the primary motor cortex (M1) is also considered a key element for dyskinesias expression, since modifications in M1 activity and the corticostriatal pathway plasticity, particularly that linked to the dopaminergic transmission in the striatum, have been found both in patients [14] and in animal models of PD [15,16,17].

Until now, dyskinesia management remains unsatisfactory. Nonetheless, available information suggests that the motor cortex may be crucial for the amelioration of this condition. In this regard, studies have shown that transcranial magnetic stimulation (TMS) improves motor symptoms and dyskinesia in PD patients, and it has been proposed as a non-invasive therapeutic option [18,19,20]. TMS is technique that uses a brief, high-intensity magnetic field generated by an electric current passing through a coil to excite or inhibit specific brain areas [21].

Growing evidence indicates that the improvement of dyskinesia symptoms with TMS might be due to a modulation of the dopaminergic system and cortical activity regularization [22,23]. Accordingly, it has been shown that subthalamic stimulation leads to the regularization of the pathological hyperactivity of M1, alleviating the motor symptoms in a PD animal model [24]. However, the understanding of the cellular and molecular mechanisms underlying TMS therapies remains limited.

To assess whether a mechanism of TMS involves the modulation of cellular activation in dyskinesias, we evaluated the expression of FosB and c-Fos in the striatum and in D1R-containing cells in the M1 and the striatum of dyskinetic animals compared with naïve (i.e., intact) rats. Furthermore, we analyzed the effect of TMS on dyskinetic behavior along with its histological marker (FosB) in hemiparkinsonian rats. In addition, we detailed the effect of TMS suspension and its reinstatement on these parameters.

## 2. Materials and Methods

### 2.1. Animals

In total, 142 male Wistar rats (230–250 g) from the breeding colony of the Instituto de Fisiología Celular, Universidad Nacional Autónoma de México (UNAM), were used. The animals were kept on a 12 h light/dark cycle (lights on at 7:00 h), at a room temperature of 22 ± 2 °C, in a cage of 3–5 subjects with food and water ad libitum. All experimental procedures were approved by the local ethical committee of the UNAM with the authorization code: RDC14-14 (28 August 2014). All the experiments were conducted in accordance with EC Directive 86/609/EEC and the Code of Practice for the Housing and Care of Animals Used in Scientific Procedures [25]. The study complied with the ARRIVE guidelines and efforts were taken to minimize the animals’ suffering throughout all experimental procedures, including daily behavior monitoring and postoperative care, to minimize discomfort and pain. In addition, no subject died before meeting the criteria for euthanasia. The strategy for identifying each subject involved assigning a numbered tag earring, accompanied by a digital database and a physical logbook for accurate tracking.

### 2.2. 6-Hydroxydopamine Lesion and Rotational Behavior

Anesthetized animals (xylazine/ketamine 10 and 90 mg/kg i.p., respectively, PiSA, Hgo., Mexico City, Mexico, Cat# Q-7833-099 and Q-7833-028) were injected with 6-hydroxydopamine (6-OHDA, Sigma-Aldrich, St. Louis, MO, USA, Cat# H4381) into the left substantia nigra pars compacta (SNc) (4.7 mm AP, 1.6 mm ML, 8.2 mm DV) [26]. Each infusion consisted of 40 µg of 6-OHDA stabilized with 32 µg/µL ascorbate in 0.5 µL of saline at 0.125 µL/min [27,28]. Two weeks following the surgery, rotational behavior was induced by an i.p. injection of apomorphine (4 mg/kg, Sigma-Aldrich, Cat# A4393) and assessed as described previously [29]. Only animals with ≥300 contralateral turns in 90 min were considered correctly lesioned and thus included in this study.

### 2.3. L-DOPA Administration and Dyskinesias Evaluation

Two weeks after the rotational behavior evaluation, daily L-DOPA administration treatment began. Two doses of L-DOPA were used: 8 mg/kg and 12 mg/kg i.p. (Sigma-Aldrich, Cat# D9628). Depending on the experimental procedure (see below), a DOPA decarboxylase inhibitor was co-administrated with L-DOPA: in experiment 1, animals received carbidopa 2 mg/kg, while in experiment 2, benserazide 15 mg/kg was administered. Dyskinesias were quantified through an abnormal involuntary movements (AIMs) scale by two experimenters, with one of them blind to the experimental group (no significant differences were found between the two qualifications). We used the basic AIMs scale, and an amplitude scale as previously described [12,30]. Only the rats that developed AIMs (128/160) were selected and randomly allocated in three groups: TMS, mock, and dyskinetic (see below). The sample sizes were determined according to the formula proposed by Charan & Kantharia (2013) [31]. The randomization method was implemented with Matlab software R2018b (MathWorks, Inc., Natick, MA, USA), using the randperm(num_subjects) function, which generates a random permutation of integers corresponding to the number of subjects, ensuring an unbiased allocation across experimental groups.

### 2.4. Transcranial Magnetic Stimulation

Animals were placed in acrylic cages designed to keep them immobile while receiving the magnetic stimulation to their heads. In this immobilization cage, the rats showed no signs of discomfort (e.g., agitation) and could not change position. TMS was generated with a pair of Helmholtz coils (Dhan 1000; Magnetoterapia, Mexico City, Mexico). Each coil consisted of 1000 turns of enameled copper wire (7 cm diameter) in plastic containers (10.5 × 10.5 × 3 cm). The stimulation consisted of an oscillatory magnetic field with extremely low-frequency magnetic fields, at 60 Hz, and a magnetic flux density of 7 mT [32]. The cages with each subject were placed 0.5 m apart, with a 30 min habituation period in an isolated room maintained at a controlled temperature of 22 °C. This was done daily between 8:00–10:00 and 14:00–16:00 h. The parameters used by this research group have been shown to improve the motor behavior of PD [33], Huntington’s disease [34] and depression on preclinical models [35]. The two coils were located dorsally and ventrally to the head. The distance between each coil and the midpoint of the head was approximately 6 cm. The magnetic field background level was <60 µT. For this study, animals were divided into the following groups:-TMS: animals were kept immobile with TMS;-Mock: animals were kept immobile with the coils turned off;-Control/dyskinetic: freely moving animals without stimulation.

### 2.5. Motor Tests

Motor performance was assessed in all rats using the Beam test and the Rotarod test before the 6-OHDA lesion, to ensure a similar motivational and motor ability at the start of the experiment, as they would be tested several more times throughout the different treatments (see Figure 1A top). During the Beam test, rats were trained to cross an increasingly thinner 2 m-long beam (12, 6, and 3 mm width) at a 15° inclination, from the bottom to the top, to reach their home cage in a maximum time of 120 s (for a detailed description see ref. [28]). The time taken to cross the beam was recorded. Those rats who did not complete the task (4/37) were excluded from this study. The Beam test was designed to assess the balance and motor coordination of rodents, and it is sensitive to nigrostriatal degeneration [36]. In the Rotarod test, rats were placed on a rotating rod that accelerated from 4 to 40 rpm over 300 s, and the time that the rats lasted on the rod before falling was logged. This test measures general motor coordination and gait [37]. The accelerating version of the task requires the animal to continually adapt its gait in response to the changing speed of the rod.

Since the AIMs manifestation prevented the animals from holding onto the beam or the rod, the dyskinetic rats were evaluated without the administration of L-DOPA. All behavioral tests were conducted between 8:00 and 12:00 h with a 30 min acclimation period. At least two animals from each group (TMS, mock, and control) were evaluated simultaneously to ensure consistency.

### 2.6. Immunohistochemistry

We performed FosB immunohistochemistry to analyze this molecular marker of dyskinesias and TH expression to verify the 6-OHDA lesion. Additionally, the samples used for the FosB histochemistry were processed for nuclei quantification with Nissl staining. To assess the cellular activation in D1R-containing cells, we carried out double-labeling immunofluorescence for c-Fos and D1 receptors.

For all the histological analysis, rats were deeply anaesthetized with sodium pentobarbital i.p. (100 mg/kg, PiSA, Cat# Q-7833-215) and intracardially perfused with 500 mL/kg phosphate buffer solution (PBS, 0.1 M) followed by 4% paraformaldehyde (PFA). Brains were extracted, post-fixed with 4% PFA, and sectioned. Brain slices were blocked for 1 h with PBS/5% bovine serum (BSA, Sigma-Aldrich, Cat# A-7030), 0.3% Triton X-100 (Sigma-Aldrich, Cat# 9002-93-1), and then incubated overnight at 4° with the primary antibody either for tyrosine hydroxylase (TH) (1:1000, rabbit polyclonal, Millipore, Billerica, MA, USA, RRID: AB_10000323), FosB (1:50 rabbit polyclonal, Santa Cruz Biotechnology, Inc., Santa Cruz, CA, USA, RRID: AB_640583), c-Fos (1:50, goat polyclonal, Santa Cruz Biotechnology, RRID: AB_2629503), or dopamine D1 receptor (1:1000 rabbit polyclonal, Abcam, Cambridge, UK, RRID:AB_445306). For c-Fos/D1R immunofluorescence, sections were incubated with the secondary antibody (1:500, 647 anti-rabbit RRID: AB_2340625 or 549 anti-goat RRID: AB_2339525, Jackson Immuno Research, West Grove, PA, USA) followed by DAPI (1:10,000, Invitrogen, Waltham, MA, USA, RRID: AB_2629482). For FosB and TH histochemistry, sections were incubated with the secondary biotinylated antibody (1:250, anti-rabbit, Millipore, RRID: AB_916366) followed by avidin-biotinylated horseradish peroxidase (Elite Kit Vector Labs, Newark, CA, USA, RRID: AB_2336819) and DAB (Sigma-Aldrich, Cat# D5637). The Nissl staining was used with 2% cresyl violet staining solution.

### 2.7. Immunohistochemistry Quantification

Analyses were determined in serial rostro–caudal coronal sections of 4–6 animals per group. FosB and Nissl staining quantification in the striatum were carried out in four sections (0.7 to −0.26 mm AP to bregma) per animal and three counting frames (dorsal, dorsolateral, and lateral) per section (0.08 mm^2^ per frame) [11]; micrographs were obtained with a Leica DM6000 microscope (40× magnification). For TH quantification in the striatum (1 to −0.26 mm AP to bregma) and SNc (−4.8 to −5.6 mm AP to bregma), three to four sections were used per animal and images were obtained with a Leica EZ4D stereomicroscope [27]. For striatum c-Fos/D1R analysis, four sections (0.7 to −0.26 mm AP to bregma) were employed per animal and four photomicrographs were acquired per area (dorsal and dorsolateral) (0.13 mm^2^ per area) with a Leica SP5 confocal microscope equipped with a 40× objective. For c-Fos/D1R quantification in the M1, three micrographs were taken with a Leica DM6000 epifluorescence microscope (40× magnification) of layer II-III, V, and VI (0.08 mm^2^ per frame) of four sections (2.2 to 1.6 mm AP to bregma) [38]. The immunofluorescence pictures were obtained with a Zeiss LSM800 confocal microscope (40× and 63× magnification; Carl Zeiss AG, Oberkochen, Germany) of the same preparations for quantification analyses. Counts were performed in processed 8-bit images, then thresholded at a standardized gray-scale level, empirically determined to allow the detection of stained nuclei/cells from a low to high intensity, with the suppression of lightly stained nuclei/cells [11]. For all images, excluding the D1R signal, threshold objects were automatically counted on both hemispheres using ImageJ 1.52 (NIH, Bethesda, MD, USA) and Matlab software (MathWorks, Inc., Natick, MA, USA) [38]. Manual count comparisons with random images were used to verify the accuracy of these methods. The quantification of D1R was manual. Positive cells were considered when D1R staining delimited DAPI staining and this region of interest showed a different intensity of the background determined by LAS-AF software. Since the c-Fos signal is mainly nuclear, the same restrictions were applied for c-Fos/D1R-positive cells [39]. Percentages of D1R+ cells in the striatum and M1 were calculated relative to DAPI-positive cells. Analyses of the c-Fos signal were carried out in the lesioned hemisphere of dyskinetic rats, comparing the left hemisphere of naïve rats as a control.

### 2.8. Statistical Analysis

The analysis was carried out using the GraphPad Prism software (GraphPad Software, LLC, San Diego, CA, USA). According to the results of the Shapiro–Wilk normality test, the data were analyzed using parametric tests. Two-way ANOVAs followed by the Tukey’s test were used for the analysis of dyskinesia scores, immunohistochemistry assays, and motor tests. All differences were considered statistically significant at *p* < 0.05.

### 2.9. Experiment 1: Effect of TMS on Dyskinetic Behavior, FosB Expression, and Motor Performance

Before TMS treatment, we evaluated the AIMs three times (2nd, 5th, and 8th week). TMS involves a high rate of ‘dose-failure’ for therapeutic applications [40], leading us to conduct two stimulation protocols. These protocols were applied in independent groups: The first protocol consisted of a 4 h stimulation period (2 h in the morning and 2 h in the afternoon) for 21 days (*n* = 9 to 14 per group; Figure 1A top), while in the second protocol, TMS was applied for only 2 h in the morning for 10 days (*n* = 9 to 11 per group; Figure 1A bottom). One day before and one day after each TMS treatment, AIMs were evaluated. The effectiveness of TMS may endure for a few days [41]; thus, part of our standardization process consisted of the evaluation of dyskinesias every two days until the efficacy was lost. This evaluation revealed that the efficacy could endure for seven days. Therefore, we investigated whether a second series of stimulation would attain the same efficacy as the first. Consequently, the stimulation was stopped for one week and then followed by a second stimulation period for each protocol. Thus, AIMs were evaluated one week following the TMS withdrawal and one day after a second series of TMS (see Figure 1A). Animals were sacrificed 2 h after the last L-DOPA injection for histological analysis, including FosB, TH expression, and Nissl staining (21-day protocol: 15th week; 10-day protocol: 12th week).

In addition, to examine in detail the dyskinetic behavior during the TMS withdrawal, we performed an AIMs evaluation every two days in an independent group of rats (Figure 2A). On the 6th day of TMS withdrawal, animals were sacrificed 2 h after the last injection of L-DOPA to analyze the FosB expression. In this group, motor tests were done two days before TMS (8th week), after 21 days of TMS (11th week), and one week after the last TMS session (12th week).

### 2.10. Experiment 2: Effect of TMS on Cellular Activation in the M1 and Striatum

Since TMS may have variable effects on the intact brain, we used non-lesioned animals without L-DOPA administration (naïve) as an intact control group to compare it with the dyskinetic rats. For the last group, AIMs were evaluated eight times before the TMS during three weeks of daily L-DOPA administration (Figure 4A; *n* = 4 or 5 per group). Then, dyskinetic and naïve animals were stimulated for 4 h for 21 days. Naïve animals were deeply anesthetized 2 h after the last stimulation for histological analysis. A day after the TMS, the last AIMs evaluation was carried out and the animals were sacrificed 2 h after the L-DOPA injection (6th week; see Figure 4A). Cellular activation was analyzed through c-Fos expression on the D1R+ cells.

## 3. Results

### 3.1. TMS Attenuated Dyskinesias Along with FosB Expression in the DA-Denervated Striatum

In experiment 1, the evaluations of the AIMs at the 2nd, 5th, and 8th week indicated the establishment of dyskinesias (Figure 1B,C). After 21 days of TMS, the AIMs score significantly decreased compared to the control groups’ scores at the 11th week (*p* < 0.01, TMS vs. Dys and Mock, Figure 1B, Appendix A). When the TMS was suspended, the AIMs score increased to a value similar to that observed in control animals (12th week, *p* > 0.95, TMS vs. Mock and Dys). After the second TMS treatment, dyskinesias significantly decreased again compared to the dyskinetic and mock groups at week 15th (*p* < 0.05 TMS vs. Dys; *p* < 0.01 TMS vs. Mock). In the second protocol, following 10 days of stimulation, the TMS group had reduced dyskinesias compared to the control groups at the 9th and 12th week (*p* < 0.05 TMS vs. Mock and Dys; Figure 1C). When the stimulation was stopped, dyskinesias augmented in this group (10th week, *p* > 0.99 TMS vs. Mock and Dys). In addition, the percentage change between the LID score prior to treatment and the score after 21 days of stimulation was greater (−60 ± 7%) than that obtained after 10 days of treatment (−46 ± 6%). The FosB analysis showed that the stimulated rats had a significant diminished expression in the DA-denervated striatum (Figure 1D) compared to controls (*p* < 0.001 TMS 10d and TMS 21d vs. Dys and Mock, Figure 1E). The expression of FosB between the DA-denervated and the intact hemisphere of stimulated rats was similar (*p* = 0.9 TMS 21d, *p* = 0.56 TMS 10d), contrasting with what was observed in the control rats (*p* < 0.01 Dys and Mock). Additionally, Nissl staining demonstrated similitude among the number of nuclei in the striatum of all animals (Appendix A). In hemiparkinsonian animals, the administration of saline solution did not induce dyskinetic behavior nor FosB expression (Appendix A).

### 3.2. TMS Withdrawal Aggravated Dyskinesias and Enhanced FosB Expression

In order to have a closer look at the effects of one week of stimulation withdrawal, we analyzed dyskinetic behavior every 2 days and the FosB expression in the striatum at the end of that week (Figure 2A). When the TMS was suspended, an increase in dyskinesias score was observed in the TMS group compared to the control animals, replicating the previous result (Figure 2B). In the first 4 days, the AIMs score of the stimulated rats was significantly lower than the control group (day 1: *p* < 0.001 TMS vs. Mock, *p* < 0.04 TMS vs. Dys; day 3: *p* < 0.01 TMS vs. Mock, *p* < 0.001 TMS vs. Dys). After 5 and 7 days without treatment, the TMS group reached the control animal values. By the end of the week, there was an exacerbated FosB expression (Figure 2C) demonstrated by a large number of FosB+ cells on the DA-denervated striatum of all rats (*p* > 0.2, Figure 2D). Furthermore, the number of positive cells between the hemispheres of the TMS group differed significantly (*p* < 0.01 denervated vs. intact striatum, Figure 2D).

### 3.3. Motor Performance Improved After TMS

In the beam test, rats required a similar time to cross all beams in the pre-treatment evaluation (Figure 3A; *p* > 0.9). After 21 days of stimulation, the TMS group had a significantly diminished time taken to cross the 6 and 12 mm beams compared to the control groups (6 mm *p* < 0.04 TMS vs. Mock and Dys; 12 mm *p* < 0.04 TMS vs. Mock). One week after the stimulation withdrawal, the treated rats’ time taken to cross increased to similar values of the control animals. Conversely, the time taken by the control animals did not differ between the evaluations. In the rotarod, none of the groups showed significant differences among tests (*p* > 0.8, Figure 3B).

We next analyzed the TH immunoreactivity in the SNc and striatum, in dyskinetic and naïve rats. The 6-OHDA neurotoxin reduced the number of TH-positive cells on the lesioned SNc (*p* < 0.001 Dys vs. naïve, Figure 3C,D) and the optical density of the DA-denervated striatum (*p* < 0.01 Dys vs. naïve, Figure 3E) of all dyskinetic rats compared to naïve animals, regardless of the TMS treatment.

### 3.4. TMS Halted c-Fos Expression in D1R-Containing Cells of the M1 and Striatum

In experiment 2, the L-DOPA administration induced dyskinesias, whereas 3 weeks of TMS significantly decreased the AIMs score (*p* = 0.0014 TMS vs. Dys, *p* = 0.0027 TMS vs. Mock; Appendix A). Histological analysis showed that there was ~30% of D1R+ cells in the M1 and ~50% in the striatum of the left hemisphere of the dyskinetic and naïve rats (Figure 4). In addition, ~80% of the c-Fos-positive nuclei had surrounding D1R immunoreactivity in both the striatum and M1 of all animals (Figure 4). In dyskinesias, c-Fos expression in the D1R+ cells was intensified in M1 of the control groups (Figure 4B–D). Conversely, the number of c-Fos/D1R+ cells in the TMS group was 50.6 ± 2% lower compared to untreated animals, and this difference was significant relative to the dyskinetic control rats (*p* = 0.0407 TMS vs. Dys, Figure 4E). Although, it was not statistically different from the mock group. In naïve animals, the number of cells that expressed c-Fos and D1R in M1 decreased by 43.4 ± 2.2% by TMS (Figure 4F–H), although this effect was not statistically significant (Figure 4I). In the striatum, c-Fos was enhanced in the dyskinetic control groups (Figure 4J,K). TMS significantly halted 78.59 ± 0.6% c-Fos expression compared to the mock and control dyskinetic groups in the D1R+ cells of the DA-depleted striatum (*p* < 0.02, Figure 4L,M). Furthermore, the c-Fos immunoreactivity was minimal in the striatum of the naïve animals (Figure 4N–P) and the TMS application had no significant effect (*p* > 0.9, Figure 4Q).

## 4. Discussion

The present study shows that TMS decreased c-Fos expression in the D1R+ cells of the DA-denervated M1 and striatum, attenuated dyskinesias along with its histological marker, and improved motor execution in hemiparkinsonian rats. In agreement to our results of TMS’s effects on dyskinetic manifestation, it has been demonstrated that magnetic exposure decreases D1R agonist-induced behaviors in 6-OHDA lesioned rats [42] and naïve animals [43]. Unlike these studies, we also evaluated whether this dyskinetic attenuation was stable, by suspending the stimulation for one week and then reinitiating it. This evaluation demonstrated that TMS had a beneficial effect in dyskinesias without developing tolerance. Nevertheless, this effect was transitory and dependent on the number of sessions. In correspondence with dyskinetic behavior, TMS downregulated the histological marker FosB and one week of the stimulation suspension promoted its overexpression in the DA-denervated striatum.

Some treatments relieve dyskinesias but also have detrimental motor effects, so it is important to assess both motor performance and dyskinesia scores [44]. Our results showed that TMS decreased dyskinesias without affecting normal motor performance, on the contrary, it induced a recovery of motor coordination as evaluated in the beam test, in agreement with previous reports [45]. Moreover, the beam test has been shown to discriminate fine motor deficits related to nigrostriatal disfunctions [36]. In contrast, in the rotarod test, we were unable to detect any effect of TMS. This may be explained by the sensitivity of the test as the rotarod assesses gross motor deficits [37]. At the cellular level, it has been suggested that motor improvement by TMS is related to dopaminergic neuronal survival via the upregulation of neurotrophic/growth factors in early parkinsonism [45,46]. However, our analysis showed that TH immunoreactivity in the SNc and striatum was similar between TMS-treated and control dyskinetic rats; this was probably due to the advanced phase of parkinsonism in which we applied TMS, since the neuronal survival induced by this type of stimulation has only been observed in initial stages [23]. Furthermore, magnetic stimulation has been shown to induce a transient increase in DA levels in the striatum [34,47], thus probably improving motor execution [34].

The attenuation of dyskinetic behavior and striatal FosB expression by TMS could indicate that this intervention disrupted the activation of D1R-containing cells, since D1R blockade abolishes FosB signaling [11] and the reduction in FosB transcriptional activity mitigates the severity of dyskinesias [10,48]. Since TMS has been shown to decrease cortical excitability [49], the reduction in dyskinetic manifestations and FosB expression in the striatum could depend on the normalization of M1 excitability, considering that striatal activity is modulated by cortical afferents [34,45,46,47,49], e.g., M1 [19]. In the present study, we also examined if TMS could affect the activity of the D1R-containing cells in these two key structures for dyskinetic behavior, M1 and the dorsal striatum, by c-Fos immunofluorescence signaling. Our results showed that TMS significantly halted the activation of D1R+ cells (78%) in the striatum and M1 (50%). Although the stimulated group did not differ statistically from the mock group, the number of c-Fos/D1R+ cells in this cortical region was significantly lower compared to the control dyskinetic rats. Importantly, the c-Fos signal in the D1R+ cells was decreased in the striatum of stimulated dyskinetic animals in contrast to both control groups. Consistent with this, magnetic stimulation has been shown to reduce c-Fos immunoreactivity in the cortex and striatum, triggered by a D1R agonist [22]. Thus, TMS regularization of cellular activity in M1 and the striatum possibly influenced dyskinetic manifestations. Moreover, D1R antagonists locally administrated to M1 reduced dyskinetic behavior [50], suggesting that cortical D1R inhibition in M1 by TMS may be a viable therapeutic mechanism.

Overall, our results show that TMS diminishes the activity of D1R+ cells in M1 and striatum, leading to dyskinesias mitigation. It is possible that TMS directly depressed the excitability of the D1R+ cells in M1, decreasing their stimulatory input on the striatum along the corticostriatal pathway. Consistent with this idea, it has been suggested that one of the beneficial mechanisms of deep brain stimulation in PD is the normalization of pathological hyperactivity in the motor cortex [24]. In addition, optogenetic stimulation in the cerebellum regularized the aberrant neuronal discharge in the cerebellar nuclei and the motor cortex, controlling FosB overexpression and attenuating dyskinesias [51].

Another possibility would be that TMS downregulated DA receptors thus inducing an increase in DA levels, which in turn regulates cellular excitability, synaptic transmission, plasticity, protein trafficking, and gene transcription [52]. In this work, we analyzed only the D1R, as it has a clearer role in dyskinesias. Future research should investigate the role of DA D2 receptors in the effects of TMS.

Our results are consistent with proposals that the effects of TMS on motor control are due to the regulation of the striatal dopaminergic system. However, we did not examine other important structures involved in dyskinesias, such as the cerebellum [20]. Taking into account that we applied non-specific stimulation to the whole brain, we cannot rule out the effects of TMS in other regions. Furthermore, animal studies using TMS for Parkinson’s disease face several limitations, including variability in protocols, differences in disease models, inconsistent outcome measures, incomplete mechanistic understanding, and challenges in translating findings to humans [53,54,55]. Addressing these limitations is essential for advancing TMS as a viable treatment option for PD.

## 5. Conclusions

The current study showed that TMS depressed FosB and c-Fos expression in D1R+ cells of the dorsal striatum and motor cortex, suggesting a mechanism by which TMS may directly mitigate dyskinesias. These results are in line with the previous evidence of the modulation of the dopaminergic system by TMS. However, questions remain such as the involvement of D2 receptors and other brain regions in the effects of TMS. Additionally, our observations highlight the therapeutic effect of TMS on dyskinesias in a PD model. Although this effect was transient and dependent on the number of stimulation sessions, it could be administrated repeatedly without generating tolerance. Further research with targeted stimulation on specific brain areas and using different animal models of PD is warranted to assess the clinical application of TMS.

## Figures and Tables

**Figure 1 brainsci-14-01214-f001:**
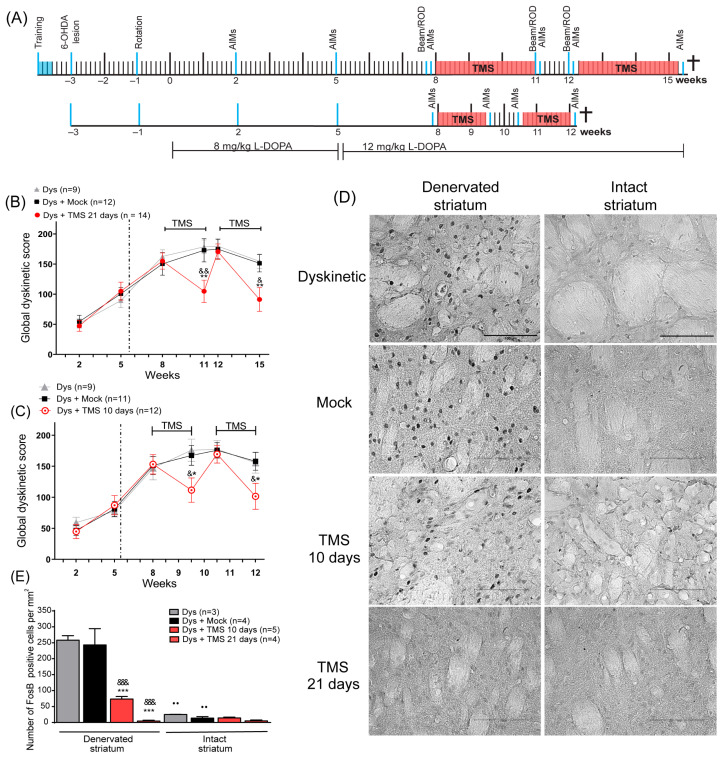
TMS significantly decreased dyskinesias and its histological marker in the dorsal striatum. (**A**) The protocols of 10 and 21 days of stimulation in experiment 1 (see Methods). All manipulations are indicated by a blue line (Rotarod test: ROD). Dyskinesia global score in rats with (**B**) 21 days and (**C**) 10 days of TMS (Dys + TMS) compared to mock (Dys + Mock) and control animals (Dys) in the 6 evaluations. ** *p* < 0.01, * *p* < 0.05 vs. Dys + Mock, && *p* < 0.01, & *p* < 0.05 vs. Dys, mean ± SEM. (**D**) Representative FosB immunoreactivity in the DA-denervated and intact striatum after 10 and 21 days of TMS, mock, and control dyskinetic animals. Scale bar = 100 µm. (**E**) Number of FosB-positive cells per mm^2^ in the DA-denervated and intact striatum in the same dyskinetic groups *** *p* < 0.001 vs. Dys + Mock, &&& *p* < 0.001 vs. Dys, ●● *p* < 0.01 DA-denervated vs. intact striatum, mean ± SEM.

**Figure 2 brainsci-14-01214-f002:**
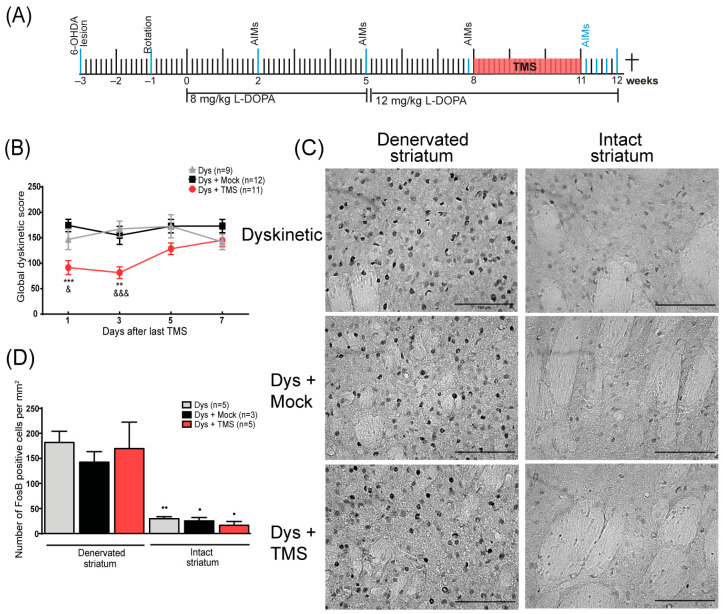
TMS withdrawal aggravated dyskinesias and enhanced FosB expression in the striatum. (**A**) The experimental protocol of the TMS withdrawal experiment. All events are indicated by a blue line. (**B**) Dyskinesia global score days after treatment withdrawal in rats with TMS (Dys + TMS) compared to mock (Dys + Mock) and control (Dys) groups. ** *p* < 0.01, *** *p* < 0.001 vs. Dys + Mock, &&& *p* < 0.001, & *p* < 0.05 vs. Dys, mean ± SEM. (**C**) Representative FosB immunohistochemistry in the DA-denervated and non-lesioned striatum after one week of stimulation retirement in dyskinesias with TMS, mock, and control groups. Scale bar = 100 µm. (**D**) Quantitative analysis of FosB-positive cells in the DA-denervated and intact striatum in dyskinetic subjects after treatment retirement. ●● *p* < 0.01, ● *p* < 0.05 DA-denervated vs. intact striatum, mean ± SEM.

**Figure 3 brainsci-14-01214-f003:**
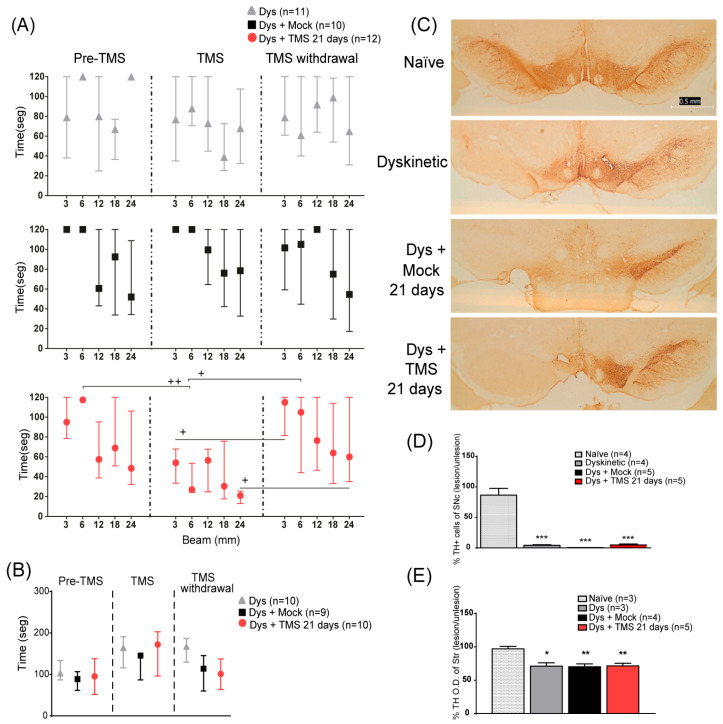
TMS improved motor performance without modifying TH expression in the SNc or the striatum. (**A**) Total time to cross 3, 6, and 12 mm beams in dyskinetic rats with 21 days of TMS (Dys + TMS 21 days, top) compared to mock (Dys + Mock, middle) and control groups (Dys, bottom) in the three evaluations (separated with dot lines). + *p* < 0.05 TMS vs. TMS withdrawal, ++ *p* < 0.01 Pre-TMS vs. TMS mean ± SEM. (**B**) Latency time to fall from the rod in dyskinetic rats in three evaluations parallel to the beam test. Mean ± SEM. (**C**) Representative TH expression of the somas on the SNc in dyskinetic rats with TMS, mock, control, and naïve rats. Scale bar = 0.5 mm. (**D**) Percentage of difference of TH-positive cells on the lesioned SNc with respect to intact SNc in dyskinetic with TMS, mock, control, and naïve groups *** *p* < 0.001 vs. naïve group, mean ± SEM. (**E**) Percentage of difference in optical density of the DA-denervated striatum regarding the intact hemisphere in the same groups as the SNc. * *p* < 0.05, ** *p* < 0.01 vs. naïve group, mean ± SEM.

**Figure 4 brainsci-14-01214-f004:**
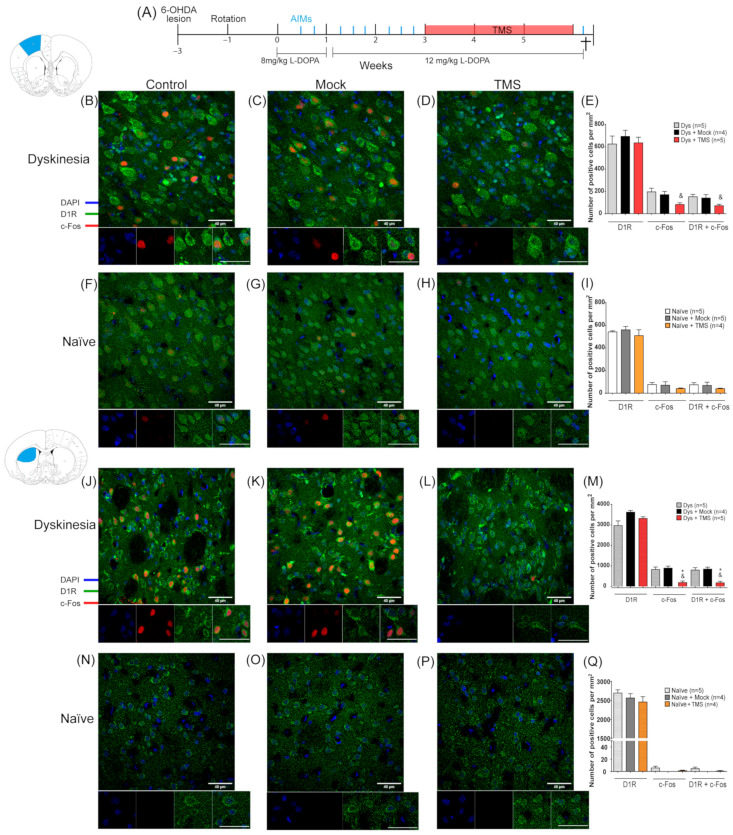
TMS depressed c-Fos signal on the dorsal striatum and motor cortex of dyskinetic animals. (**A**) The experimental protocol of experiment 2 (see Methods). AIMs evaluations are indicated by a blue line. Representative c-Fos and DR1 immunostaining on the left (DA-denervated) M1 (**B**–**D**,**F**–**H**) and the left striatum (**J**–**L**,**N**–**P**) along with its quantitative analysis of positive cells per mm^2^ (**E**,**I**,**M**,**Q**) in dyskinetic and naïve animals with or without TMS treatment. Scale bar = 40 µm. * *p* <0.05 vs. Dys + Mock, & *p* < 0.05 vs. Dys, n.s. not significant, mean ± SEM.

## Data Availability

The original data are held in a public repository at https://doi.org/10.6084/m9.figshare.24986016.v1.

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
