# Peer review of "Transcranial Magnetic Stimulation Attenuates Dyskinesias and FosB and c-Fos Expression in a Parkinson’s Disease Model"

_brainsci, 2024, doi:10.3390/brainsci14121214_

Round 1

Reviewer 1 Report

Comments and Suggestions for Authors

First of all, thank you for giving me the opportunity to review this very important and interesting manuscript.

The topic is very relevant, since there are few works devoted to experimental confirmation of the effectiveness of TMS in the treatment of patients with dyskinesia, which develops against the background of long-term use of dopamine in the treatment of Parkinson's disease.

The title matches the content.

Define dyskinesia please.

Add a link from recent clinical studies supporting this information “Several studies have suggested a major role for dopamine D1 receptors (D1R) in dyskinesia manifestation”

Suggested that FosB and ΔFosB are present in the nuclei of TIDA neurons, as would be expected of transcription factors responding to neurotoxic insult. Please add more information about FosB and c-Fos expression in Parkinson's disease. do they happen normally? is there a useful function? When were they first identified? found only in experimental models? Are they found in other diseases?

In purpose give a description of naive rats. Not all readers will understand this. (Sham rats).

Materials and methods are described in detail in all sections:

·         2.1. Animals

·         2.2 6-hydroxydopamine lesion and rotational behavior

·         2.3. L-DOPA administration and dyskinesias evaluation

·         2.4. Transcranial magnetic stimulation

·         2.5. Motor tests

·         2.6. Immunohistochemistry

·         2.7. Immunohistochemistry quantification

However, the organization of the text is very poor. difficult to read. Try to present the information in these sections point by point.

In line 201: “There were no exclusions for the data analysis”. Did all experimental animals survive striatal denervation? Have animal models of Parkinson's disease been successful in all cases? Please indicate in the text.

Add design for this research

What method was used to determine the sample size?

Add figures for Experiment 1 and experiment 2. This makes the experiments easier for readers to understand.

Results The results were demonstrated to be not very clear. The authors indicated only the reliability of the increase or decrease in value, while the dynamics of these changes in percentage were not indicated. Comparative analyzes in percentages between the study groups were not carried out. Please add more accurate descriptions. In addition, demonstrate the results point by point. Very difficult to read.

Congratulations to the authors for Figures 1-4, which are very informative and make it easier to understand the dynamics of change during the experiment.

Discussion and conclusions follow logically from the results of the study and are fully consistent with the purpose of the study.

During the discussion, the authors presented their results and compared them with the results of other authors. In addition, the authors provided explanations and put forward some explanatory hypotheses:

·         TCM can induce a transient increase in DA levels in the striatum thus probably improving motor execution.

·         TMS can decrease cortical excitability

·         TMS significantly halted the activation of D1R+ cells (78%) in the striatum and M1

·         TMS diminishes the activity of D1R+ cells in M1 and striatum leading to dyskinesias mitigation.

·         TMS contributes to normalization of pathological hyperactivity in the motor cortex.

·         TMS downregulated DA receptors thus inducing an increase of DA levels, which in turn regulates cellular excitability, synaptic transmission, plasticity, protein trafficking and gene transcription.

Author Response

First of all, thank you for giving me the opportunity to review this very important and interesting manuscript.

The topic is very relevant, since there are few works devoted to experimental confirmation of the effectiveness of TMS in the treatment of patients with dyskinesia, which develops against the background of long-term use of dopamine in the treatment of Parkinson's disease.

The title matches the content.

Comments 1. Define dyskinesia please.

Response 1: Thank you for requesting clarification on dyskinesia. We have added the requested information in lines 57 to 64, which further elaborates on this topic. We hope this provides the necessary context and clarity.

Comments 2. Add a link from recent clinical studies supporting this information “Several studies have suggested a major role for dopamine D1 receptors (D1R) in dyskinesia manifestation”

Response 2: Thank you for your suggestion. We have added a paragraph between lines 66 and 72 to support the statement regarding the role of dopamine D1 receptors (D1R) in the manifestation of dyskinesia, including references to recent clinical studies.

Comments 3. Suggested that FosB and ΔFosB are present in the nuclei of TIDA neurons, as would be expected of transcription factors responding to neurotoxic insult. Please add more information about FosB and c-Fos expression in Parkinson's disease. do they happen normally? is there a useful function? When were they first identified? found only in experimental models? Are they found in other diseases?

Response 3: We appreciate your interest in the normal function of FosB/FosB-like transcription factors and their expression in this study. While some of this information is already provided in lines 72 to 76, we have gladly expanded and enriched these details further in lines 77 to 81.

Comments 4. In purpose give a description of naive rats. Not all readers will understand this. (Sham rats).

Response 4: Indeed, the reviewer is right. Naïve (intact) rats were used only for histological comparisons, not for behavioral ones, as intact rats do not show any motor impairment at all. We have added the specification “(i.e., intact)” in the paragraph in question to avoid confusion (line 104).

Comments 5. Materials and methods are described in detail in all sections:

  • 2.1. Animals
  • 2.2 6-hydroxydopamine lesion and rotational behavior
  • 2.3. L-DOPA administration and dyskinesias evaluation
  • 2.4. Transcranial magnetic stimulation
  • 2.5. Motor tests
  • 2.6. Immunohistochemistry
  • 2.7. Immunohistochemistry quantification

However, the organization of the text is very poor. difficult to read. Try to present the information in these sections point by point.

Response 5: The reviewer is right; we detected some omissions and inaccuracies in the presentation of our methods. We have followed the reviewer's suggestions and edited the Methods section, in particular points 2.9 and 2.10 in which we specify the procedures followed for each protocol (10-day TMS and 21-day TMS).

Comments 6. In line 201: “There were no exclusions for the data analysis”. Did all experimental animals survive striatal denervation? Have animal models of Parkinson's disease been successful in all cases? Please indicate in the text.

Response 6: The statement “There were no exclusions for the data analysis” refered to the fact that there were no exclusion criteria during the statistical analysis, however that phrase is confusing and we decided to remove it. In fact, there were exclusions in each of the experimental phases as detailed in the text (line 246).

Comments 7. Add design for this research

Response: The experimental designs are described in points 2.8 and 2.9 of the Methods section and Figs. 1A, 2A and 4A of the Results section.

Comments 8. What method was used to determine the sample size?

Response 8: The sample size was calculated according to the formula proposed by Charan & Kantharia (2013).

Comments 8.1 Add figures for Experiment 1 and experiment 2. This makes the experiments easier for readers to understand.

Response 8.1: As mentioned above, the figures describing Experiments 1 and 2 can be found in the Results section (Fig. 1A, 2A, and 4A). We decided to include them in this section rather than in the Methods because this way it is easier to understand the results included in the respective figures, i.e., Figs. 1, 2 and 4. We believe that with the new wording of Experiments 1 and 2 and the repeated references to these figures, the readers will find both the methods and results sections easier to appreciate.

Results

Comments 9. The results were demonstrated to be not very clear. The authors indicated only the reliability of the increase or decrease in value, while the dynamics of these changes in percentage were not indicated. Comparative analyzes in percentages between the study groups were not carried out. Please add more accurate descriptions. In addition, demonstrate the results point by point. Very difficult to read.

Congratulations to the authors for Figures 1-4, which are very informative and make it easier to understand the dynamics of change during the experiment.

Response 9: We appreciate your observations regarding the clarity of the results. We realize that we did not clearly explain how the percentage change was calculated. In response, we have reviewed the results and modified lines 297 to 300 to include more detailed descriptions and provide the comparative analysis in percentages between the study groups. We hope this revision improves the clarity of the results.

Comments 10. Discussion and conclusions follow logically from the results of the study and are fully consistent with the purpose of the study. 

During the discussion, the authors presented their results and compared them with the results of other authors. In addition, the authors provided explanations and put forward some explanatory hypotheses: 

  • TCM can induce a transient increase in DA levels in the striatum thus probably improving motor execution.
  • TMS can decrease cortical excitability
  • TMS significantly halted the activation of D1R+ cells (78%) in the striatum and M1
  • TMS diminishes the activity of D1R+ cells in M1 and striatum leading to dyskinesias mitigation.
  • TMS contributes to normalization of pathological hyperactivity in the motor cortex.
  • TMS downregulated DA receptors thus inducing an increase of DA levels, which in turn regulates cellular excitability, synaptic transmission, plasticity, protein trafficking and gene transcription.

Response 10: We’re pleased that you found the discussion and conclusions to be logical and aligned with the study’s objectives. We appreciate your recognition of our efforts to compare our findings with existing literature and to propose explanatory hypotheses. Your comments are greatly motivating and affirm our approach to presenting and interpreting the results.

Submission Date

30 October 2024

Date of this review

10 Nov 2024 11:41:51

Reviewer 2 Report

Comments and Suggestions for Authors

 The Authors experimentally attempted to verify the mechanism of the usefulness of transcranial magnetic stimulation (TMS) as a non-invasive therapy for Parkinson's disease symptoms attenuation, motor symptoms, and dyskinesias.  They investigated whether TMS regulates cellular activation, with c-Fos immunoreactivity, on dopamine D1 receptor-positive cells of the striatum and motor cortex of dyskinetic (n=14) and naïve rats (n=14). They also evaluated the TMS effect on dyskinesias global score along with its molecular marker, FosB, in the striatum (n=67). They revealed that transcranial magnetic stimulation reduces c-Fos expression in dopamine D1 receptor-positive cells in the striatum and motor cortex. TMS significantly attenuated dyskinesias, an effect that was accompanied by a low expression of FosB in the striatum. The Authors concluded that TMS depressed FosB and c-Fos expression in D1R+ cells of the dorsal striatum and motor cortex, suggesting a mechanism by which TMS may directly mitigate dyskinesias.

The scientific value of the work is significant, the research model is designed appropriately for the study's purposes, the research methods are precise enough to achieve the intended goals. The number of experimental animals is small (N=28), which is a limitation of the study.

1. The Introduction chapter provides a sufficient overview of the state of art in the field of PD, both in terms of basic research and clinical trials, emphasising therapeutic attempts of dyskinesia treatment. The Authors cite two relevant papers (12, 13) on TMS utilization in PD treatment and activation of dopamine D1-like receptors in mice (16). I would recommend similarly well formulating the study's objectives in the Abstract as it has been presented at the end of the Introduction section.

2. The Materials and Methods section precisely describes inducing PD in rats and its treatment. Motor and behavioural tests and immunohistochemistry procedures are clearly described. I would recommend that Authors more precisely define the principles of TMS procedures. This subsection is economical in its wording, and it would be difficult to repeat the experiment in another laboratory based on the description provided.

3. I would like to express my appreciation to the Authors for the precise, clear and graphically perfect presentation of the Results of the work, which I read with great pleasure.

4.  A significant advantage of the Discussion is its attempt to explain the mechanism of action of TMS in alleviating the symptoms of Parkinson's disease.  Nevertheless, mentioning the study limitations (including the number of animals) is difficult to find anywhere in the Discussion, which should have been mentioned by the Authors.

5. The conclusions correspond to the formulated objectives of the work.

6. I did not detect any errors in citing, listing, or selecting references. 

Author Response

Comments and Suggestions for Authors

 The Authors experimentally attempted to verify the mechanism of the usefulness of transcranial magnetic stimulation (TMS) as a non-invasive therapy for Parkinson's disease symptoms attenuation, motor symptoms, and dyskinesias.  They investigated whether TMS regulates cellular activation, with c-Fos immunoreactivity, on dopamine D1 receptor-positive cells of the striatum and motor cortex of dyskinetic (n=14) and naïve rats (n=14). They also evaluated the TMS effect on dyskinesias global score along with its molecular marker, FosB, in the striatum (n=67). They revealed that transcranial magnetic stimulation reduces c-Fos expression in dopamine D1 receptor-positive cells in the striatum and motor cortex. TMS significantly attenuated dyskinesias, an effect that was accompanied by a low expression of FosB in the striatum. The Authors concluded that TMS depressed FosB and c-Fos expression in D1R+ cells of the dorsal striatum and motor cortex, suggesting a mechanism by which TMS may directly mitigate dyskinesias.

The scientific value of the work is significant, the research model is designed appropriately for the study's purposes, the research methods are precise enough to achieve the intended goals. The number of experimental animals is small (N=28), which is a limitation of the study.

Comments 1. The Introduction chapter provides a sufficient overview of the state of art in the field of PD, both in terms of basic research and clinical trials, emphasising therapeutic attempts of dyskinesia treatment. The Authors cite two relevant papers (12, 13) on TMS utilization in PD treatment and activation of dopamine D1-like receptors in mice (16). I would recommend similarly well formulating the study's objectives in the Abstract as it has been presented at the end of the Introduction section.

Response 1: Following the reviewer's recommendation, we have rewritten the objectives in the Abstract for greater clarity.

Comments 2. The Materials and Methods section precisely describes inducing PD in rats and its treatment. Motor and behavioural tests and immunohistochemistry procedures are clearly described. I would recommend that Authors more precisely define the principles of TMS procedures. This subsection is economical in its wording, and it would be difficult to repeat the experiment in another laboratory based on the description provided.

Response 2: Thank you for your thoughtful feedback. We appreciate your observations regarding the TMS procedures. In response, we have expanded the Introduction (lines 90-92) and Materials and Methods section (lines 154-157) to provide a more detailed explanation of the principles of TMS and ensuring that the procedure is clearly defined and reproducible in other laboratories. We hope this revision addresses your concerns and enhances the clarity of our methodology.

Comments 3. I would like to express my appreciation to the Authors for the precise, clear and graphically perfect presentation of the Results of the work, which I read with great pleasure.

Response 3: We’re delighted to hear that you found the presentation of our results clear, precise, and visually effective. Your appreciation is truly encouraging, and we’re pleased that the figures contributed to a positive reading experience.

Comments 4. A significant advantage of the Discussion is its attempt to explain the mechanism of action of TMS in alleviating the symptoms of Parkinson's disease.  Nevertheless, mentioning the study limitations (including the number of animals) is difficult to find anywhere in the Discussion, which should have been mentioned by the Authors.

Response 4: Thank you for your insightful feedback. We appreciate your suggestion to include a discussion of the study limitations. In response, we have added information regarding the limitations, including the number of animals, in lines 458 to 466. We hope this addition provides a more balanced perspective and enhances the overall clarity of the discussion.

Comments 5. The conclusions correspond to the formulated objectives of the work.

Comments 6. I did not detect any errors in citing, listing, or selecting references.